# Generation of Photonic Nanojet Using Gold Film Dielectric Microdisk Structure

**DOI:** 10.3390/ma16083146

**Published:** 2023-04-16

**Authors:** Xintao Zeng, Ning Su, Weiming Zhang, Zhibin Ye, Pinghui Wu, Bin Liu

**Affiliations:** 1Research Center for Photonic Technology, Fujian Provincial Key Laboratory for Advanced Micro-Nano Photonics Technology and Devices, Quanzhou 362000, China; 2Key Laboratory of Information Functional Material for Fujian Higher Education, Quanzhou Normal University, Quanzhou 362000, China; 3College of Electrical and Information Engineering, Quzhou University, Quzhou 324000, China; 4Rural Revitalization Institute, Linyi University, Linyi 276000, China

**Keywords:** photonic nanojet (PNJ), gold-film dielectric microdisk, finite difference time domain (FDTD), surface plasmon polariton (SPP)

## Abstract

Due to their narrow beam waist size, high intensity, and long propagation distance, photonic nanojets (PNJs) can be used in various fields such as nanoparticle sensing, optical subwavelength detection, and optical data storage. In this paper, we report a strategy to realize an SPP-PNJ by exciting a surface plasmon polariton (SPP) on a gold-film dielectric microdisk. In detail, an SPP is excited by the grating–coupling method, then it irradiates the dielectric microdisk to form an SPP-PNJ. The characteristics of the SPP-PNJ, including maximum intensity, full width at half maximum (FWHM), and propagation distance, are studied by using finite difference time domain (FDTD) numerical solutions. The results demonstrate that the proposed structure can produce a high-quality SPP-PNJ, the maximum quality factor of which is 62.20, and the propagation distance of the SPP-PNJ is 3.08 λ. Furthermore, the properties of the SPP-PNJ can be modified flexibly by changing the thickness and refractive index of the dielectric microdisk.

## 1. Introduction

A photonic nanojet (PNJ) is a narrow-intensity electromagnetic beam emerging from the shadow-side surface of a plane-wave-illuminated dielectric microcylinder or microsphere (with a diameter greater than the illuminating wavelength) propagating into the surrounding medium [1]. Since the first report on PNJs by Benincasa et al. in 1987 [2], and since the term “PNJ” was introduced by Chen et al. in 2004 [3], they have attracted much attention from researchers for their generation method, transmission characteristics, and potential applications [4,5,6,7,8,9,10,11,12,13,14,15,16,17,18]. Nowadays, PNJs are widely used in applications including nanoparticle sensing, all-optical switches, optical sub-wavelength detection, optical data storage, super-resolution imaging, and orbital angular momentum [19,20,21,22,23,24,25,26]. For instance, Patrick et al. reported the direct experimental observation of photonic nanojets created by single latex microspheres illuminated by a plane wave at a wavelength of 520 nm in 2008 [27], Shen et al. proposed an ultralong PNJ generated by two-layer dielectric microspheres [28], and Sundaram et al. proposed a nanoscale high-intensity light-focusing method using pure dielectric aspherical scatterers in 2014 [29]. Gu et al. used liquid-filled microspheres to generate an ultralong PNJ in 2015 [30], Victor Pacheco-Pena et al. reported a method of generating PNJs based on the periodic arrangement of a 3D dielectric cuboid micro-structure in 2016 [31], and Geints et al. presented a systematic study of the key characteristics (field intensity enhancements, spatial extents) of 2D and 3D photonic nanojets (PNJs) produced by geometrically regular micron-sized dielectric particles illuminated by a plane laser wave in 2018 [32]. Recently, we obtained an ultrahigh-quality factor PNJ from truncated microtoroid structures [33], and Xing et al. investigated a curved truncated dielectric microcylinder structure using the finite element method, which can form an ultralong PNJ with the longest effective length, in 2022 [34].

The key properties of the PNJ, including maximum intensity, the propagating distance of the PNJ, and full width at half maximum (FWHM), depend on the geometric parameters (size, shape, etc.), refractive indices of the micro-particles, and the output wavelength of the incident light. Traditionally, the main method of generating PNJs is irradiating dielectric microparticles such as microspheres, microcylinders, and micro-cuboids. However, microspheres, microcylinders, and micro-cuboids have shortcomings such as inconvenience in experimental operations. To overcome these drawbacks, McCloskey et al. proposed a new scheme to generate SPP-PNJs using silicon nitride dielectric microdisks in 2012 [35]. Due to the limited field portion of SPPs in metals, they are often difficult to transmit over long distances, which seriously limits their application value, especially in planar photonic devices. For example, D. Ju et al. first reported nanojet effects excited by a surface plasmon polariton on the shadow-side surfaces of dielectric microdisks positioned on gold films in 2013. Additionally, in 2020, I. Minin first reported the experimental demonstration of plasmonic nanojet production by a micro-cuboid deposited on a gold film.

In this paper, we propose a structure to realize an SPP-PNJ by exciting a surface plasmon polariton (SPP) on a gold-film dielectric microdisk. In detail, by using the finite difference time domain (FDTD) method, we attempted to adjust the characteristics of an SPP-PNJ by adjusting the refractive index and thickness of the microdisk. The influences of the refractive index and thickness of the dielectric microdisk on the maximum intensity, full width at half maximum, propagation distance, and quality factor of the SPP-PNJ were analyzed in detail. The results show that the propagation distance of the SPP-PNJ is 3.08 λ. The characteristics of the SPP-PNJ are flexibly tunable by changing the thickness and refractive index of the dielectric microdisk. In addition, an SPP-PNJ with a quality factor of 62.20 can be achieved. Compared with the dielectric microsphere, microcylinder, and micro-cuboid, the dielectric microdisk structure can excite the long propagation distance of the SPP-PNJ, and the characteristics of the SPP-PNJ can be flexibly tunable. Therefore, the proposed dielectric microdisk can provide a new idea for further expanding the generation path of SPP-PNJs and conveniently tuning the characteristics of SPP-PNJs.

## 2. Structure of Gold-Film Dielectric Microdisk

Here, we consider that the FDTD method based on vector electromagnetic wave theory can accurately illustrate the propagation of a light wave in dielectric media [36]. Therefore, the structure of the gold-film dielectric microdisk was simulated by using the FDTD calculation. In order to ensure the accuracy of the calculation, a non-uniform mesh with a minimum step of 5 nm was applied. Perfectly matched layers (PML) were arranged around the boundaries.

Figure 1 shows a schematic diagram of an SPP-PNJ formed by a gold-film dielectric microdisk structure. It consists of a dielectric microdisk, gold film, grating, and a dielectric substrate. An SPP was excited by the grating–coupling method, which then irradiated the dielectric microdisk with a radius *R*, thickness *h,* and refractive index *n* along the negative x direction. Here, the dielectric microdisk was deposited onto a gold film. The lower layer of the gold film was a dielectric substrate with a semi-infinite thickness. The whole structure was symmetric with respect to the y = 0 plane, and the upper surface of the gold film was located in the z = 0 plane. Assuming that the whole structure was enclosed by air, we set the wavelength of the incident wave as 800 nm, and the refractive indices of the background and dielectric substrate were set to 1.0 and 1.5, respectively. The material of the gold film and grating was set as Au (Gold) according to Johnson and Christy. According to the refractive indices of the grating and background, we set the period of the grating as 583.94 nm, which could couple the incident wave to the gold film to excite the 800 nm SPP. The radius of the dielectric microdisk was set to 3 μm.

To evaluate the light beam quality of the SPP-PNJ more comprehensively, the quality factor Q is used to measure the SPP-PNJ, which is written as [37]:(1)$$Q=\frac{I_{\max}\times L}{F}$$
where *I*_max_ is the maximum intensity of the SPP-PNJ, which represents the maximum optical intensity reached along the light propagation direction; *L* is the propagation distance of the SPP-PNJ, which represents the distance of the x-axis corresponding to the optical intensity between the maximum intensity of the SPP-PNJ and 1/e of the maximum intensity; *F* is the FWHM of the SPP-PNJ, which represents the width of the y-axis corresponding to the SPP-PNJ’s attenuation from the maximum intensity to the half maximum intensity.

According to Equation (1), *Q* depends on *I*_max_, *L*, and *F*. Obviously, a higher *Q* can be obtained by a higher *I*_max_ or *L* and a lower *F*. An SPP-PNJ with a higher *Q* is expected in many application fields.

## 3. Simulations

Figure 2 visualizes the FDTD-computed optical intensity formed by the gold-film dielectric microdisk. The optical intensity of the SPP-PNJ is shown by the color bar scale. Here, the refractive index of the dielectric microdisk is 1.5. The thickness and radius of the dielectric microdisk are 1.0 and 3.0 μm, respectively. The illuminating wave is a z-axis polarized plane wave with a wavelength *λ* = 800 nm, propagating along the y-axis with an initial amplitude of 1. The plane wave is irradiated onto the grating for coupling, resulting in plasma resonance, and the generated SPP propagates along the x-axis, producing an SPP-PNJ with different properties. Figure 2a depicts an SPP-PNJ in the x–y longitudinal cut plane. Figure 2b shows a three-dimensional surface plot of the SPP-PNJ. From this figure, we see that the gold-film dielectric microdisk structure can generate the SPP-PNJ. The generated SPP-PNJ is a narrow-intensity electromagnetic beam emerging from the shadow-side surface of the gold-film dielectric microdisk structure propagating along the x-axis.

In order to analyze the influence of the gold-film dielectric microdisk on the characteristics of SPP-PNJs, we study the influence of the dielectric microdisk’s refractive index and thickness.

### 3.1. The Characteristic Parameters of SPP-PNJs with Different Refractive Indices of the Dielectric Microdisk

Figure 3 shows the optical intensity field and the maximum intensity *I*_max_ evolution formed by different dielectric microdisk refractive indices *n* = 1.3, *n* = 1.4, *n* = 1.5, *n* = 1.6, and *n* = 1.7, respectively. The optical intensity of the SPP-PNJ is shown by the color bar scale. Here, the thickness and radius of the dielectric microdisk are *h* = 1.0 μm and *R* = 3.0 μm, respectively. The effective wavelength of the SPP is *λ* = 800.0 nm. It can be seen in Figure 3a–e that the peak of the optical intensity field shifts toward the shadow-side surface of the dielectric microdisk by decreasing the refractive index of the dielectric microdisk *n*. In addition, we can see in Figure 3f that the maximum intensity *I*_max_ increases when the value of *n* increases from 1.3 to 1.6. However, when the value of n increases from 1.6 to 1.7, *I*_max_ decreases with *n*. Hence, when *n* reaches an appropriate value, the optical intensity field peak emerges from the shadow-side surface of the dielectric microdisk and an SPP-PNJ is formed. The maximum intensity *I*_max_ of an SPP-PNJ formed by a gold-film dielectric microdisk can be tuned by designing the refractive index *n* of the dielectric microdisk.

Figure 4 shows the optical intensity distributions of the SPP-PNJ along the x-axis with different dielectric microdisk refractive indices *n*; the corresponding SPP-PNJ propagation distances *L* are shown in Figure 5. We can see in Figure 4 that the position of the maximum intensity along the y-axis is similar, at approximately −2 μm. In Figure 4 and Figure 5, it can be seen that *L* decreases when the value of *n* increases from 1.3 to 1.6, while *L* increases when the value of *n* increases from 1.6 to 1.7. The value of the maximum propagation distance *L*_max,n_ is 2.46 μm (3.08 *λ*) when the value of *n* is 1.3. The value of the minimum propagation distance *L*_min,n_ is 0.32 μm (0.40 *λ*) when the value of *n* is 1.6. Therefore, we can design the value of *n* to obtain the appropriate propagation *L* of the SPP-PNJ according to different applications.

Figure 6 shows the FWHM *F* of the SPP-PNJ with different refractive indices *n* of the dielectric microdisk. We can clearly see that *F* decreases when the value of *n* increases from 1.3 to 1.6, and *F* increases when the value of *n* increases from 1.6 to 1.7. Notably, when *n* is in the range of 1.5 to 1.7, *F* is less than 0.4 μm (0.5 *λ*), which plays an important role in super-resolution imaging. The value of *F* is 0.49 (0.61 *λ*) and 0.41 (0.51 *λ*) when the value of *n* is 1.3 and 1.4, respectively, which can be used in nanoparticle sensing and optical sub-wavelength detection. Notably, we can see that the intensity dip at y = 0 when *n* is in the range of 1.5 to 1.7. The reason for the significant decrease in intensity at y = 0 is that the strongest point of the focused beam formed by the interference between the field scattered by the medium’s exit surface and the incident field is not at the center, but rather forms the two strongest regions symmetrically about the central axis.

From the above analyses, we find that the gold-film dielectric microdisk can form an SPP-PNJ with a smaller diffraction limit, which is similar to the SPP-PNJs generated by irradiating dielectric microspheres or microcylinders with plane waves. The quality factor *Q* of an SPP-PNJ formed by a gold-film microdisk can be calculated according to Equation (1). The characteristic parameters of SPP-PNJ, including *I*_max_, *F*, *L*, and *Q* with different dielectric microdisk refractive indices, are listed in Table 1. It clearly shows that the largest quality factor *Q*_max_ is 49.47 and the smallest quality factor *Q*_min_ is 20.83, which can be achieved by setting the refractive index of the dielectric microdisk to 1.5 and 1.6, respectively. Therefore, the appropriate SPP-PNJ for each application can be obtained by designing the refractive index of the dielectric microdisk.

### 3.2. The Characteristic Parameters of SPP-PNJs with Different Thicknesses of the Dielectric Microdisk

According to the above analysis, the maximum value of *Q* of an SPP-PNJ can be realized when *n* = 1.5. Therefore, the thickness of the dielectric microdisk is analyzed with n set to 1.5.

The optical intensity field and the maximum intensity *I*_max_ of SPP-PNJs for different thicknesses of dielectric microdisks are shown in Figure 7. We can see the SPP-PNJs formed by dielectric microdisks with different thicknesses with a radius *R* = 3.0 μm and refractive index *n* = 1.5. The color bar scale demonstrates the intensity of the SPP-PNJ. Figure 7 shows that the maximum intensity *I*_max_ increases when the value of *h* increases from 0.6 to 0.8 μm. On the contrary, when the value of n increases from 0.8 to 1.0, *I*_max_ decreases with *h*. We can find that the peak of the optical intensity field shifts toward the shadow-side surface of the dielectric microdisk by increasing the refractive index of the dielectric microdisk *h* from 0.6 to 0.8 μm.

To analyze the characteristic parameters of the SPP-PNJ, Figure 8 shows the optical intensity distributions of the SPP-PNJ along the y-axis with different dielectric microdisk refractive indices *n*; the corresponding SPP-PNJ propagation distances *L* are shown in Figure 9. We can clearly see that the position of maximum intensity along the y-axis is −2.3 μm when *h* is 0.8 μm. However, the attenuation of the SPP-PNJ is faster. Furthermore, when the *h* is 0.9 μm, the position of maximum intensity along the y-axis is −2 μm and the attenuation of the SPP-PNJ is slower. Therefore, in Figure 8 and Figure 9, we can see that *L* decreases when the value of *h* increases from 0.6 to 0.8 μm. When *h* is 0.9 μm, *L* increases suddenly and then decreases continuously. The value of the maximum propagation distance *L*_max,h_ is 1.72 μm (2.15 *λ*) when the value of *h* is 0.9 μm. The value of the minimum propagation distance *L*_min,h_ is 0.76 (0.95 *λ*) when the value of *h* is 0.7 μm. Obviously, the value of the maximum propagation distance *L*_max,h_ (2.15 *λ*) is shorter than the value of *L*_max,n_ (3.08 *λ*). Therefore, we can design *n* and *h* to obtain the appropriate propagation *L* of the SPP-PNJ according to different applications.

Figure 10 shows the full width at half maximum *F* of the SPP-PNJ with different thicknesses of *h* of the dielectric microdisk. It can be seen that when *h* increases from 0.6 μm to 0.8 μm, *F* increases with *h*. However, while *h* increases from 0.8 to 1.0 μm, *F* decreases with *h*. It is noteworthy that *F* is always less than 0.4 μm (0.5 *λ*) when *h* is in the range of 0.6 to 1.0 μm. This means that the incident wave can form an SPP-PNJ with a smaller diffraction limit by designing the appropriate value of *h*.

Table 2 shows the characteristic parameters of the SPP-PNJ with different thicknesses *h* of the dielectric microdisk. It can be seen in Table 2 that when *h* = 0.8 μm and the *I*_max_ of the SPP-PNJ is 14.33 times stronger than that of the exciting light, *F* = 0.37 μm (0.46 *λ*) and *L* = 0.98 μm (1.23 *λ*). The largest obtained value of *Q* for the SPP-PNJ was 62.20. The smallest value of *Q* was 23.92, which can be realized when *h* = 0.7 μm. This will benefit the application of SPP-PNJs in super-resolution optical imaging. This indicates that with the increase of *h*, *I*_max_ and *F* increase first and then decrease. However, the relationship between *h* and the propagation distance *L* of the SPP-PNJ is more complicated.

According to Table 1 and Table 2, the relationship between *Q*, *n*, and *h* is shown in Figure 11. It can be seen in Figure 10 that *Q* is greatly different with different *h* and *n* values, respectively. However, it is found that the maximum value of *Q* can be achieved by setting the thickness of the dielectric microdisk *h*. For example, when the value of *h* is 0.9 μm, *Q* reaches a maximum value of 62.20, and when the value of *n* is 1.5, *Q* reaches a maximum value of 49.47. However, compared with the influence of different thicknesses of *h*, the longest propagation distance *L* can be realized by different refractive indices. When the value of *n* is 1.3, *L* reaches a maximum value of 2.46 μm (3.08 *λ*). Therefore, the most suitable SPP-PNJ can be achieved by adjusting the refractive index and thickness of the dielectric microdisk in practical applications.

## 4. Conclusions

In this paper, we study the characteristics of an SPP-PNJ generated by a gold-film dielectric microdisk structure. In detail, the effects of the dielectric microdisk’s refractive index and thickness on the maximum intensity, full width at half maximum, propagation distance, and quality factor of the SPP-PNJ are analyzed. The results show that the gold-film dielectric microdisk structure can generate high-quality SPP-PNJs, which are similar to the SPP-PNJs generated by irradiating dielectric microspheres or microcylinders with plane waves. In addition, compared with dielectric microspheres or microcylinders, the gold-film dielectric microdisk structure can adjust the characteristics of the SPP-PNJ by changing the refractive index and thickness of the dielectric microdisk. Therefore, the proposed dielectric microdisk can provide new ideas for further expanding the generation path of SPP-PNJs and conveniently tuning their characteristics.

## Figures and Tables

**Figure 1 materials-16-03146-f001:**
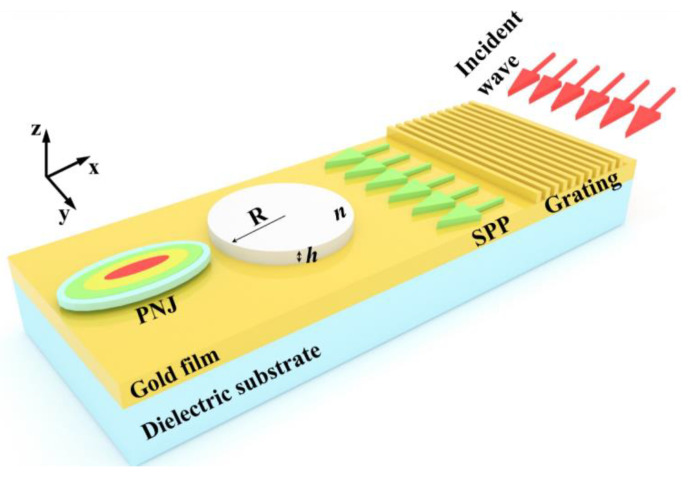
Schematic diagram of a PJ formed by a gold-film dielectric microdisk structure.

**Figure 2 materials-16-03146-f002:**
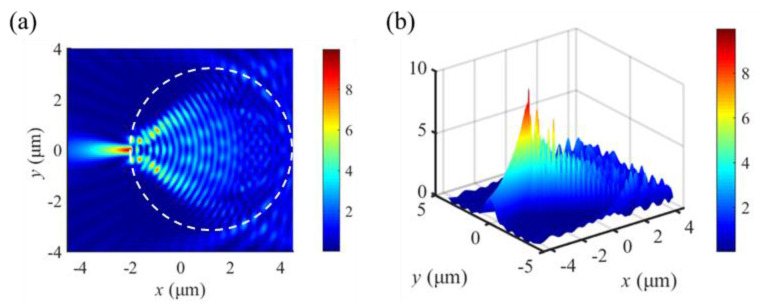
FDTD-computed optical field intensity formed by different dielectric microdisks. (**a**) Two-dimensional (2D) simulation results of intensity distributions on the x–y plane, (**b**) three-dimensional (3D) simulation results for the intensity distributions.

**Figure 3 materials-16-03146-f003:**
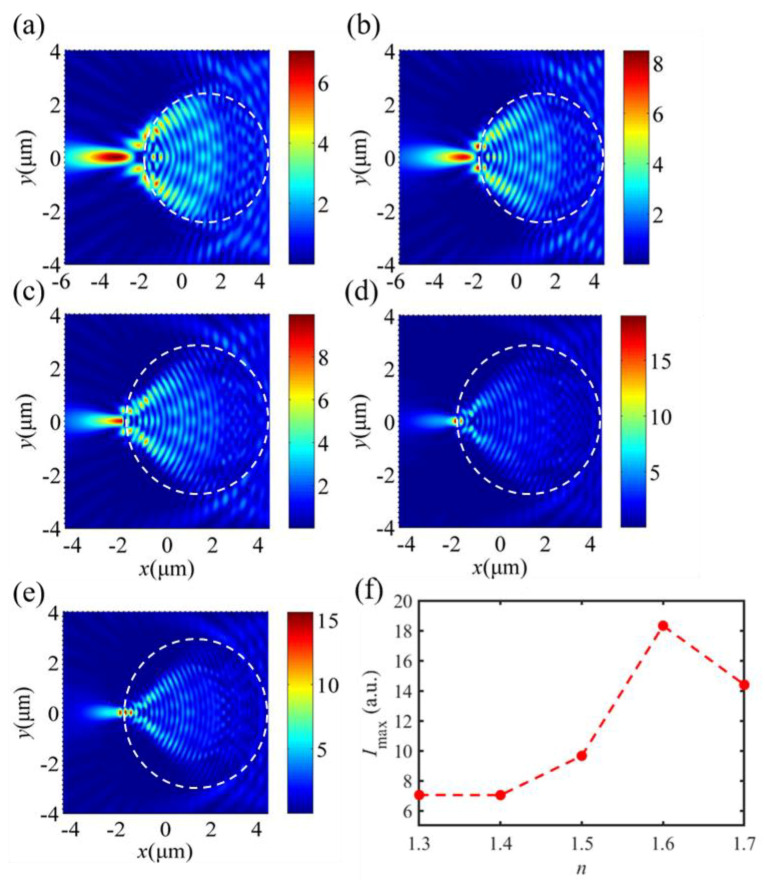
Optical field intensity evolution of the SPP-PNJ formed by different dielectric microdisk refractive indices: (**a**) *n* = 1.3, (**b**) *n* = 1.4, (**c**) *n* = 1.5, (**d**) *n* = 1.6, (**e**) *n* = 1.7, and (**f**) the maximum intensity *I*_max_ evolution with different refractive indices of the dielectric microdisk.

**Figure 4 materials-16-03146-f004:**
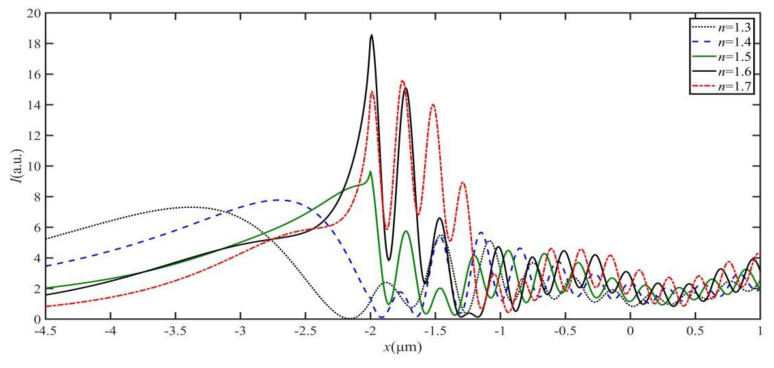
Optical intensity distributions of the SPP-PNJ along the y-axis with different dielectric microdisk refractive indices *n*.

**Figure 5 materials-16-03146-f005:**
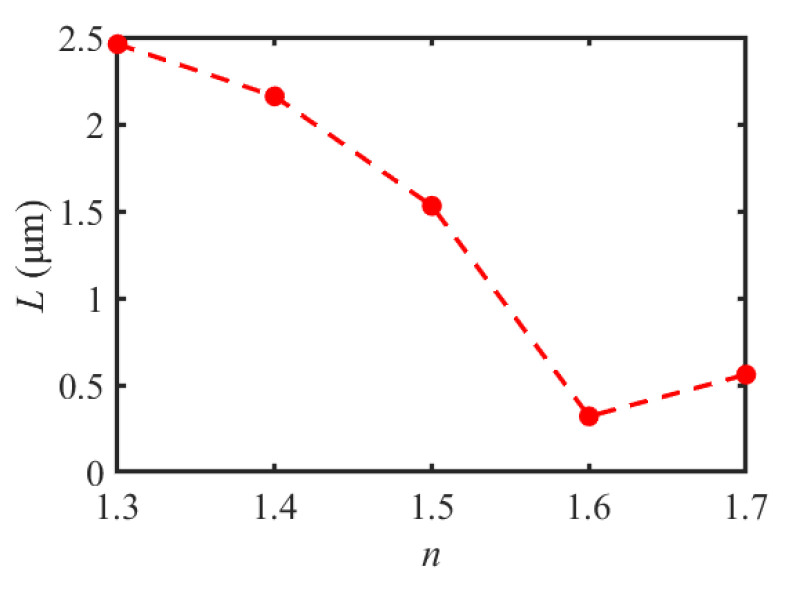
Comparison of the propagation distance *L* of the SPP-PNJ with different dielectric microdisk refractive indices *n*.

**Figure 6 materials-16-03146-f006:**
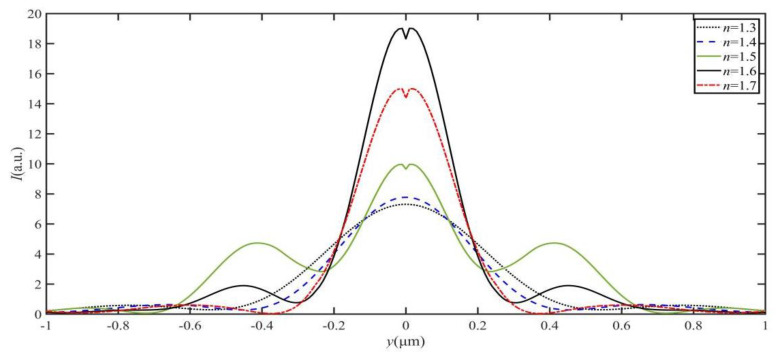
Comparison of full width at half maximum *F* of the SPP-PNJ with different dielectric microdisk refractive indices *n*.

**Figure 7 materials-16-03146-f007:**
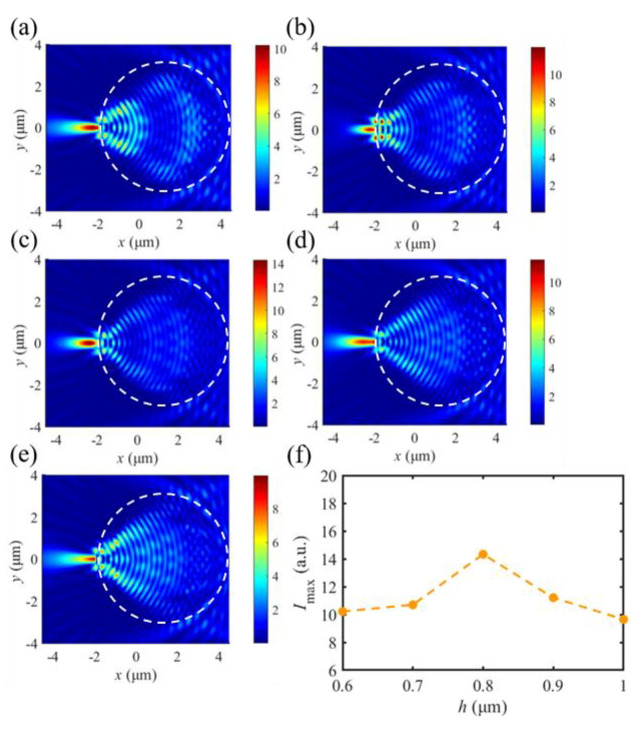
Optical field intensity evolution of the SPP-PNJ formed by different thicknesses of the dielectric microdisk: (**a**) *h* = 0.6 μm, (**b**) *h* = 0.7 μm, (**c**) *h* = 0.8 μm, (**d**) *h* = 0.9 μm, (**e**) *h* = 1.0 μm, and (**f**) the maximum intensity *I*_max_ evolution with different thicknesses of the dielectric microdisk.

**Figure 8 materials-16-03146-f008:**
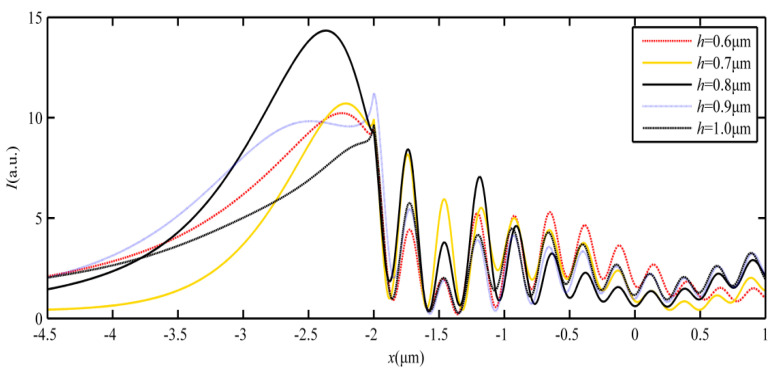
Optical intensity distributions of the SPP-PNJ along the y-axis with different thicknesses *h* of the dielectric microdisk.

**Figure 9 materials-16-03146-f009:**
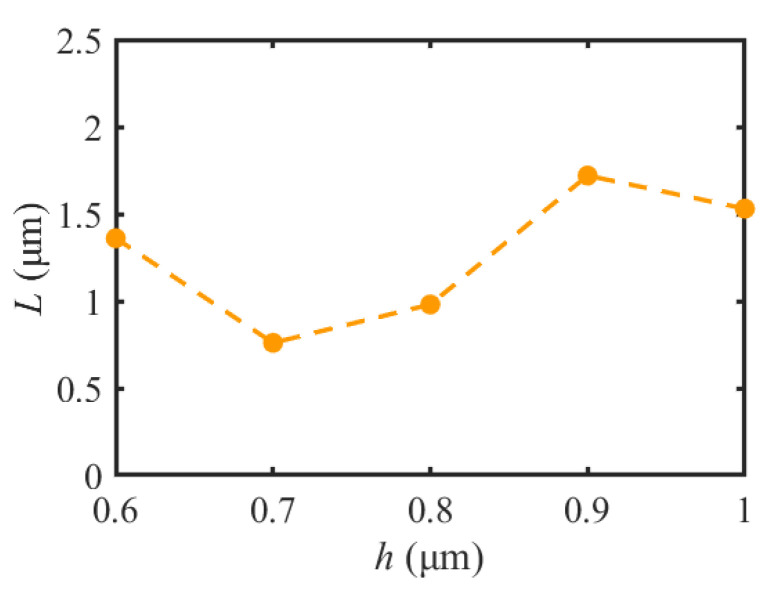
Comparison of the propagation distance *L* of the SPP-PNJ with different thicknesses *h* of the dielectric microdisk.

**Figure 10 materials-16-03146-f010:**
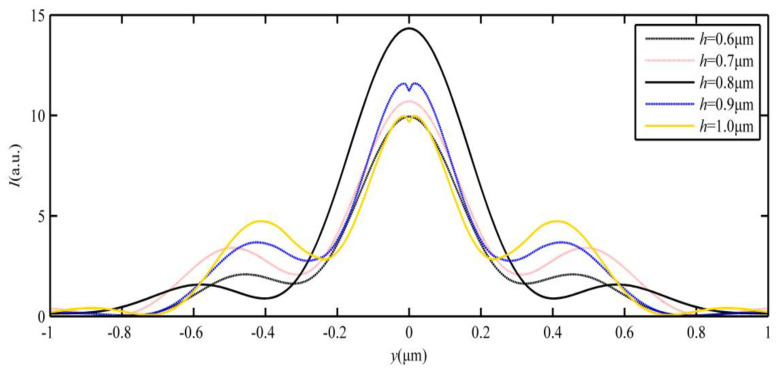
Comparison of full width at half maximum *F* of the SPP-PNJ with different thicknesses *h* of the dielectric microdisk.

**Figure 11 materials-16-03146-f011:**
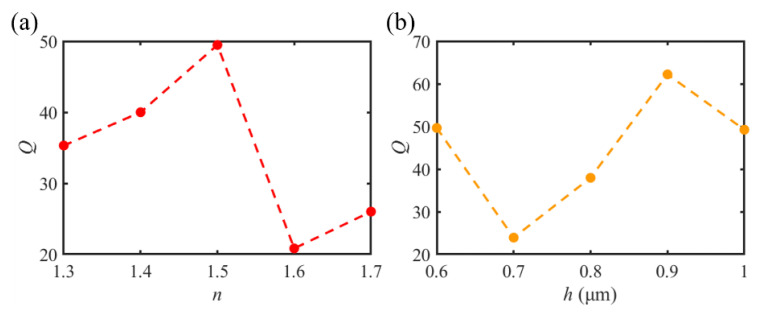
Quality factor *Q* for (**a**) different dielectric microdisk thickness *n* and (**b**) different dielectric microdisk refractive index *h*.

**Table 1 materials-16-03146-t001:** Characteristic parameters of an SPP-PNJ excited with different refractive indices *n* of the dielectric microdisk.

*N*	*I_max_* (*a.u.*)	*F* (μm)	*L* (μm)	*Q*
1.3	7.04	0.49 (0.61 λ)	2.46 (3.08 λ)	35.31
1.4	7.53	0.41 (0.51 λ)	2.16 (2.69 λ)	40.00
1.5	9.65	0.30 (0.37 λ)	1.53 (1.91 λ)	49.47
1.6	18.32	0.28 (0.35 λ)	0.32 (0.40 λ)	20.83
1.7	14.38	0.31 (0.38 λ)	0.56 (0.70 λ)	26.01

**Table 2 materials-16-03146-t002:** Characteristic parameters of the SPP-PNJ excited with different thicknesses *h* of the microdisk.

*h* (μm)	*I*_max_ (*a.u.*)	*F* (μm)	*L* (μm)	*Q*
0.6	10.22	0.28 (0.35 λ)	1.36 (1.70 λ)	49.64
0.7	10.70	0.34 (0.43 λ)	0.76 (0.95 λ)	23.92
0.8	14.33	0.37 (0.46 λ)	0.98 (1.23 λ)	37.96
0.9	11.21	0.31 (0.39 λ)	1.72 (2.15 λ)	62.20
1.0	9.65	0.30 (0.38 λ)	1.53 (1.91 λ)	49.47

## Data Availability

Not applicable.

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
