# Peer review of "Generation of Photonic Nanojet Using Gold Film Dielectric Microdisk Structure"

_materials, 2023, doi:10.3390/ma16083146_

Round 1
Reviewer 1 Report
In the paper by X. Zeng et al. the photonic nanojet (PNJ) effect is theoretically considered as observed in the shadow side of wavelength-scale dielectric objects. Specifically, the authors address the PNJ parameters emerging via the surface plasmon polariton (SPP) scattering at a dielectric microdisk. SPP is excited by grating-coupling method at the gold foil-dielectric interface and then irradiates the dielectric microdisk to form a PNJ. The influence of disc refractive index (RI) and height on jet intensity, length and width is studied by means of the FDTD calculations.
Actually, the very idea of PNJ generation by the SPP illumination of a dielectric disc does not seem novel and was previously proposed by D. Ju et al. in 2013 [https://doi.org/10.1063/1.4802958]. Later, in 2020, I. Minin et al. [https://doi.org/10.1364/OL.391861] reported the first experimental demonstration of the plasmonic nanojet production by a microcuboid deposited on a gold film. Note, all these works aren’t referenced in the manuscript under review. Moreover, the authors of the above cited papers came to conclusions that there exists an optimal range of the microparticle RI (between 1.3 and 1.7) for better PNJ generation. Besides, the microdisk and microcuboid thickness effect on the plasmonic PNJ was also considered. Therefore, neither the idea nor the simulation results presented in the manuscript by X. Zeng et al. aren’t new.
Based on the abovementioned points, I cannot recommend this paper for publication in MDPI materials in its present form. Besides, I have some comments to the manuscript, which demand the authors’ reaction before the paper can be considered for future submission.
1. The work motivation is unclear for me because I don’t understand why “microsphere, microcylinder, and micro-cuboid are not convenient for experimental operation and device integration”? The proposed photonic-plasmonic microstructure is much more complicated and challenged in fabrication.
2. Refs. [3, 4] are wrong cited.
3. Some key works studying the PNJ parameters are missed, e.g., [https://doi.org/10.1364/OL.39.000582; https://doi.org/10.1364/OE.16.006930; https://doi.org/10.1088/2040-8986/aac1d9 ].
4. The finite difference formulation presented in Section 2 is incomplete because lacks the details about the micrograting coupler discretization, laser illumination specifics (wavelength in free space, inclination) and gold RI model.
5. PNJ figure of merit (quality factor in Eq.1) is first introduced not in [30] but by S.-C. Kong et al. in [https://doi.org/10.1364/OE.17.003722];
6. In 2D intensity distributions in figs. 2, 3 and 7 show the disc boundaries.
7. Which realistic optical materials correspond the RI values used in the simulations?
8. In figs. 6 and 10 one cannot see FWHM of a PNJ but only the lateral intensity profiles are plotted. What a pronounced intensity dip at y=0 corresponds to in these distributions?
9. Typos: line 127, should by “x-axis”; line 167, should be “R=3um”.
10. When presenting the summarizing graphs (fig.11) and draw the conclusions, all the possible combinations of (n,h) should be considered. Possible, some “best” combinations are missed?
11. English should be cardinally improved.
Author Response
Response to reviewer' comments:
Thank the reviewer for these constructive comments concerning our manuscript (Manuscript ID: 476325, Type: research article) entitled “Generation of Photonic Nanojet Using Gold Film Dielectric Microdisk Structure”. These comments are all valuable and very helpful for improving our paper. We have studied the comments and revised our manuscript accordingly. Our responses to the reviewer’ comments are listed as follows:
Comments and Suggestions for Authors
In the paper by X. Zeng et al. the photonic nanojet (PNJ) effect is theoretically considered as observed in the shadow side of wavelength-scale dielectric objects. Specifically, the authors address the PNJ parameters emerging via the surface plasmon polariton (SPP) scattering at a dielectric microdisk. SPP is excited by grating-coupling method at the gold film-dielectric interface and then irradiates the dielectric microdisk to form a PNJ. The influence of disk refractive index (RI) and height on jet intensity, length and width is studied by means of the FDTD calculations.
Actually, the very idea of PNJ generation by the SPP illumination of a dielectric disc does not seem novel and was previously proposed by D. Ju et al. in 2013 [https://doi.org/10.1063/1.4802958]. Later, in 2020, I. Minin et al. [https://doi.org/10.1364/OL.391861] reported the first experimental demonstration of the plasmonic nanojet production by a microcuboid deposited on a gold film. Note, all these works aren’t referenced in the manuscript under review. Moreover, the authors of the above cited papers came to conclusions that there exists an optimal range of the microparticle RI (between 1.3 and 1.7) for better PNJ generation. Besides, the microdisk and microcuboid thickness effect on the plasmonic PNJ was also considered. Therefore, neither the idea nor the simulation results presented in the manuscript by X. Zeng et al. aren’t new.
Based on the abovementioned points, I cannot recommend this paper for publication in MDPI materials in its present form. Besides, I have some comments to the manuscript, which demand the authors’ reaction before the paper can be considered for future submission.
Thank you for your positive remark and suggestions for our work.
Red Roman: The contents added in our revised version.
Reviewer 1
- The work motivation is unclear for me because I don’t understand why “microsphere, microcylinder, and micro-cuboid are not convenient for experimental operation and device integration”? The proposed photonic-plasmonic microstructure is much more complicated and challenged in fabrication.
Thank you for your suggestion. This work motivation is improve the propagation length of SPP-PNJ by exciting a surface plasmon polariton (SPP) on a gold film-dielectric microdisk and the characteristics of SPP-PNJ can be flexibly tunable by changing the thickness and refractive index of dielectric microdisk. Due to the limited field portion of SPP in metals, it is often difficult to transmit over long distances, which seriously limits its application value, especially in planar photonic devices. However, the generation of SPP-PNJ is beneficial for improving the propagation length of SPP and for coupling between photonic devices using SPP, which has potential application value in the optical field coupling of photonic devices. Therefore, we proposed a structure to realize a PNJ by exciting a surface plasmon polariton (SPP) on a gold film-dielectric microdisk. In addition, the characteristics of PNJ can be flexibly tunable by changing the thickness and refractive index of dielectric microdisk.
According to your advice, we add the statement of the motivation of our paper in revised manuscript (19-20) and (56-66): The results demonstrate that the proposed structure can produce high quality SPP-PNJ, the maximum of quality factor is 62.20, and the propagation of SPP-PNJ is 3.08λ. (19-20)
Besides, due to the limited field portion of SPP in metals, it is often difficult to transmit over long distances, which seriously limits its application value, especially in planar photonic devices.
In this paper, we propose a structure to realize a PNJ by exciting a surface plas-mon polariton (SPP) on a gold film-dielectric microdisk. In detail, by using fi-nite-difference time-domain (FDTD), we tried to adjust the characteristics of PNJ by adjusting the refractive index and thickness of the microdisk. The influences of the refractive index and thickness of the dielectric microdisk on the maximum intensity, full width at half maximum, propagation distance and quality factor of PNJ were analyzed in detail. The results show that the propagation of SPP-PNJ is 3.08λ. The characteristics of PNJ can be flexibly tunable by changing the thickness and refractive index of dielectric microdisk. In addition, a PNJ with the quality factor of 62.20 can be achieved. Compared with dielectric microsphere, microcylinder, and micro-cuboid, the dielectric microdisk structure can excite the long propagation distance SPP-PNJ, and the characteristics of PNJ can be flexibly tunable. Therefore, the proposed dielectric microdisk can provide a new idea for further expanding the generation path of SPP-PNJ and conveniently tuning the characteristics of SPP-PNJ. (56-66)
- [3, 4] are wrong cited.
Thanks for your helpful advice. The error of reference [3, 4] has been corrected in the revised manuscript.
- Some key works studying the PNJ parameters are missed, e.g., [https://doi.org/10.1364/OL.39.000582; https://doi.org/10.1364/OE.16.006930; https://doi.org/10.1088/2040-8986/aac1d9 ].
Thanks for your careful review and guidance of this paper. Missing papers on the key work of PNJ parameters have been added to the revised manuscript. Patrick et al. report the direct experimental observation of photonic nanojets created by single latex microspheres illuminated by a plane wave at a wavelength of 520 nm in 2008, and they measure nanojet sizes as small as 270 nm FWHM for a 3µm sphere at a wavelength λ of 520 nm [1]. Sundaram et al. proposed a nanoscale high intensity light focusing method using pure dielectric aspherical scatterers. Light scattering from nonspherically symmetric pure dielectric structures is examined. From the finite element full-wave analysis, it is found that eardrop-shaped scatterers can focus visible light to a ∼ 10 nm spot with an intensity enhancement ∼ 105 when the incident light is radially polarized [2]. Geints et al. present the systematic study of key characteristics (field intensity enhancement, spatial extents) of the 2D- and 3D-photonic nanojets (PNJs) produced by geometrically-regular micron-sized dielectric particles illuminated by a plane laser wave in 2018. By means of the finite-difference time-domain calculations, we highlight the differences and similarities between PNJs in these two spatial configurations for curved- (sphere, circular cylinder) and rectangle-shaped scatterers (cube, square bar) [3].
- Ferrand P, Wenger J, Devilez A, et al. Direct imaging of photonic nanojets. Opt. Express 2008, 16(10): 6930-6940.
- Sundaram V M, Wen S. Nanoscale high-intensity light focusing with pure dielectric nonspherical scatterer. Opt. Lett. 2014, 39(3): 582-585.
- Geints Y E, Zemlyanov A A, Minin O V, et al. Systematic study and comparison of photonic nanojets produced by dielectric microparticles in 2D-and 3D-spatial configurations. J OPTICS-UK 2018, 20(6): 065606.
- The finite difference formulation presented in Section 2 is incomplete because lacks the details about the micrograting coupler discretization, laser illumination specifics (wavelength in free space, inclination) and gold RI model.
Thanks for your careful review and suggestion for our paper. To completed present the finite difference formulation in section 2. We add that: we setting the wavelength of incident wave is 800nm, the refractive index of background and dielectric substrate is set to 1.0 and 1.5 respectively. The material of gold film and grating is set to Au (Gold) – Johnson and Christy. According to the refractive index of grating and background, we set the period of grating is 583.94 nm, which can couple the incident wave to the gold film to excite the 800 nm SPP. The radius of dielectric microdisk is set to 3 μm.
According to your advice, we add the statement of the micrograting coupler discretization, laser illumination specifics (wavelength in free space, inclination) and gold RI model (line 89-94): Assuming that the whole structure is enclosed by air. We setting the wavelength of incident wave is 800 nm, the refractive index of background and dielectric substrate is set to 1.0 and 1.5 respectively. The material of gold film and grating is set to Au (Gold) – Johnson and Christy. According to the refractive index of grating and background, we set the period of grating is 583.94 nm, which can couple the incident wave to the gold film to excite the 800 nm SPP. The radius of dielectric microdisk is set to 3 μm.
- PNJ figure of merit (quality factor in Eq.1) is first introduced not in [30] but by S.-C. Kong et al. in [https://doi.org/10.1364/OE.17.003722];
Thanks for your careful review and guidance of this paper. The literature cited by PNJ figure of merit has been corrected in the original text, where Li et al. first proposed the concept.
- In 2D intensity distributions in 2, 3 and 7 show the disc boundaries.
Thanks for your careful review and guidance of this paper. The unmarked microdisk boundary problem you mentioned in figs. 2, 3 and 7 has been resolved, as shown below.
Figure 1. FDTD-computed optical field intensity formed by different dielectric microdisk. (a) 2D simulations results of intensity distributions on the xy plane, (b) 3D simulations results for the intensity distributions.
Figure 2. Optical field intensity evolution of the PNJ formed by different dielectric microdisk refractive index (a) n=1.3, (b) n=1.4, (c) n=1.5, (d) n=1.6, (e) n=1.7 and (f) the maximum intensity Imax evolution with different refractive index of dielectric microdisk.
Figure 3. Optical field intensity evolution of the PNJ formed by different thickness of dielectric microdisk (a) h=0.6 μm, (b) h=0.7 μm, (c) h=0.8 μm, (d) h=0.9 μm, (e) h=1.0 μm and (f) the maximum intensity Imax evolution with different thickness of dielectric microdisk.
- Which realistic optical materials correspond the RI values used in the simulations?
- When the refractive index is 1.3, ethylene tetrafluoroethylene can be used [1].
- When the refractive index is 1.4, methyl silicone can be used [2].
- When the refractive index is 1.5, Borocrown glass can be used [3].
[1] Dominguez, I.; Corres, J.; Matias, I. R.; et al. High sensitivity lossy-mode resonance refractometer using low refractive index PFA planar waveguide. Opt Laser Technol 2023, 162: 109235.
[2] Mosley, D. W.; Khanarian, G.; Conner, D. M.; et al. High refractive index thermally stable phenoxyphenyl and phenylthiophenyl silicones for light‐emitting diode applications. JAPS 2014, 131(3).
[3] Chand, E. M.; Sharma, E. S.; Sharma, E. R. K. Demonstration of chromatic dispersion in borosilicate crown glass microstructure optical fiber. IJMER Vol, 2.
- In figs. 6 and 10 one cannot see FWHM of a PNJ but only the lateral intensity profiles are plotted. What a pronounced intensity dip at y=0 corresponds to in these distributions?
Thank you for your careful review and suggestion for our paper. The reason for the decrease in intensity at y=0 should be that the strongest point of the focused beam formed by the interference between the field scattered by the medium's exit surface and the incident field is not at the center, but rather forms two strongest regions symmetrical about the central axis.
According to your advice, we add the statement of why the decrease in intensity at y=0 (170-174): Noteworthily, we can find that the intensity dip at y=0 when n from 1.5 to 1.7. The reason for the significant decrease in intensity at y=0 should be that the strongest point of the focused beam formed by the interference between the field scattered by the me-dium's exit surface and the incident field is not at the center, but rather forms two strongest regions symmetrical about the central axis.
- Typos: line 127, should by “x-axis”; line 167, should be “R=3um”.
Thank you for your valuable suggestion. Error in line 127, has been changed to "x axis". Error in line 167, has been changed to "R=3um".
- When presenting the summarizing graphs (fig.11) and draw the conclusions, all the possible combinations of (n,h) should be considered. Possible, some “best” combinations are missed?
The main purpose of this paper is not to find out the best combination of (n,h), but to flexibly regulate the characteristics of PNJ through the transformation of (n,h). After a lot of experiments, the (n,h) combination given in Figure 11 is currently the best combination, which has a long effective length, a narrow beam waist and an ideal maximum light field intensity. The quality of PNJ generated by (n,h) in other ranges is not so good. However, we will try to find a better (n,h) combination and optimize the structure in the future.
[1] H, Yang.; Trouillon, R.; Huszka, G.; et al. Super-resolution imaging of a dielectric microsphere is governed by the waist of its photonic nanojet. NANO LETT 2016, 16(8): 4862-4870.
- English should be cardinally improved.
Thank you for your valuable and thoughtful comments. We have carefully checked and improved the English writing in the revised manuscript.

Reviewer 2 Report
The authors present a novel approach to excite NPJ trough the use of Gold Film dielectric microdisk. I found this study original and well presented, thus I recommend publication in the present form.
Author Response
hank the reviewer for these constructive comments concerning our manuscript (Manuscript ID: 476325, Type: research article) entitled “Generation of Photonic Nanojet Using Gold Film Dielectric Microdisk Structure”. These comments are all valuable and very helpful for improving our paper.
Reviewer 3 Report
The reviewed manuscript reports on a new method of obtaining photonic nanojets by excitation of a surface plasmon polariton on a gold film-dielectric micro-disk. My observations thereupon are laid out below:
1. The Authors need to refer to the following publications:
a) Horiuchi, N. Photonic nanojets. Nature Photon 6, 138–139 (2012). https://doi.org/10.1038/nphoton.2012.43
b) J. Zhu, L. Goddard. All-dielectric concentration of electromagnetic fields at the nanoscale: the role of photonic nanojets. Nanoscale Advances, Issue 12, 2019.
https://doi.org/10.1039/C9NA00430K
c) Patel, H.; Kushwaha, P.; Swami, M. Generation of highly confined photonic nanojet using crescent-shape refractive index profile in microsphere. Opt. Comm., v. 415, 140-145 (2018). https://doi.org/10.1016/j.optcom.2018.01.050
d) C. Lin, Y. Lee, C. Liu. Optimal photonic nanojet beam shaping by mesoscale dielectric dome lens. J. Appl. Phys., v. 127, 243110 (2020). https://doi.org/10.1063/5.0007611
2. The advantages of the new method are not clear, since it is similar to photonic nanojet (PNJ) generation by irradiating dielectric microspheres or microcylinders with plane waves. It is necessary to accentuate the advantages of the new method of photonic nanojet generation in comparison with other methods.
3. It should be pointed out that this work is theoretical and predictive, whereas no experimental implementation of it has so far been demonstrated. It is also necessary to specify the parameters of the exciting radiation. The manuscript only gives the wavelength (800 nm), but other key parameters need to be provided, such as the spectral width, divergence, &c.
If the Authors address the issues pointed out above in a forthcoming revision of their work, it may be published in Materials
Author Response
Response to reviewer' comments:
Thank the reviewer for these constructive comments concerning our manuscript (Manuscript ID: 476325, Type: research article) entitled “Generation of Photonic Nanojet Using Gold Film Dielectric Microdisk Structure”. These comments are all valuable and very helpful for improving our paper. We have studied the comments and revised our manuscript accordingly. Our responses to the reviewer’ comments are listed as follows:
Comments and Suggestions for Authors
In the paper by X. Zeng et al. the photonic nanojet (PNJ) effect is theoretically considered as observed in the shadow side of wavelength-scale dielectric objects. Specifically, the authors address the PNJ parameters emerging via the surface plasmon polariton (SPP) scattering at a dielectric microdisk. SPP is excited by grating-coupling method at the gold film-dielectric interface and then irradiates the dielectric microdisk to form a PNJ. The influence of disk refractive index (RI) and height on jet intensity, length and width is studied by means of the FDTD calculations.
Actually, the very idea of PNJ generation by the SPP illumination of a dielectric disc does not seem novel and was previously proposed by D. Ju et al. in 2013 [https://doi.org/10.1063/1.4802958]. Later, in 2020, I. Minin et al. [https://doi.org/10.1364/OL.391861] reported the first experimental demonstration of the plasmonic nanojet production by a microcuboid deposited on a gold film. Note, all these works aren’t referenced in the manuscript under review. Moreover, the authors of the above cited papers came to conclusions that there exists an optimal range of the microparticle RI (between 1.3 and 1.7) for better PNJ generation. Besides, the microdisk and microcuboid thickness effect on the plasmonic PNJ was also considered. Therefore, neither the idea nor the simulation results presented in the manuscript by X. Zeng et al. aren’t new.
Based on the abovementioned points, I cannot recommend this paper for publication in MDPI materials in its present form. Besides, I have some comments to the manuscript, which demand the authors’ reaction before the paper can be considered for future submission.
Thank you for your positive remark and suggestions for our work.
Red Roman: The contents added in our revised version.
Reviewer 2
- The Authors need to refer to the following publications:
- a) Horiuchi, N. Photonic nanojets. Nature Photon 6, 138–139 (2012). https://doi.org/10.1038/nphoton.2012.43
- b) J. Zhu, L. Goddard. All-dielectric concentration of electromagnetic fields at the nanoscale: the role of photonic nanojets. Nanoscale Advances, Issue 12, 2019.
https://doi.org/10.1039/C9NA00430K
- c) Patel, H.; Kushwaha, P.; Swami, M. Generation of highly confined photonic nanojet using crescent-shape refractive index profile in microsphere. Opt. Comm., v. 415, 140-145 (2018). https://doi.org/10.1016/j.optcom.2018.01.050
- d) C. Lin, Y. Lee, C. Liu. Optimal photonic nanojet beam shaping by mesoscale dielectric dome lens. J. Appl. Phys., v. 127, 243110 (2020). https://doi.org/10.1063/5.0007611
Answer: Thank you for your valuable suggestion. I have added the following references according to your request:
[1] Horiuchi, N. Photonic nanojets. Nature Photon 2012, 6(3): 138-139.
[2] J, Zhu.; Goddard, L. L. All-dielectric concentration of electromagnetic fields at the nanoscale: the role of photonic nanojets. NANOSCALE ADV 2019, 1(12): 4615-4643.
[3] Patel, H. S.; Kushwaha, P. K.; Swami, M. K. Generation of highly confined photonic nanojet using crescent-shape refractive index profile in microsphere. OPT COMMUN 2018, 415: 140-145.
[4] C. B, Lin.; Lee, Y. T.; C. Y, Liu. Optimal photonic nanojet beam shaping by mesoscale dielectric dome lens. J APPL PHYS 2020, 127(24): 243110.
- The advantages of the new method are not clear, since it is similar to photonic nanojet (PNJ) generation by irradiating dielectric microspheres or microcylinders with plane waves. It is necessary to accentuate the advantages of the new method of photonic nanojet generation in comparison with other methods.
Answer: Thanks for your helpful advice. Due to the limited field portion of SPP in metals, it is often difficult to transmit over long distances, which seriously limits its application value, especially in planar photonic devices. However, the generation of SPP-pnj PNJ is beneficial for improving the transmission length of SPP and for coupling between photonic devices using SPP, which has potential application value in the optical field coupling of photonic devices. In this paper, we report a strategy to realize a PNJ by exciting a surface plasmon polariton (SPP) on a gold film-dielectric microdisk. In detail, a SPP is excited by grating-coupling method, then it ir-radiates the dielectric microdisk to form a PNJ. We tried to adjust the characteristics of PNJ by adjusting the refractive index and thickness of the microdisk. The influences of the refractive index and thickness of the dielectric microdisk on the maximum intensity, full width at half maximum, propagation distance and quality factor of PNJ were analyzed in detail, so as to improve the quality factor of PNJ. The results show that the gold film dielectric microdisk structure can generate high-quality PNJ, which is similar to PNJ generated by irradiating dielectric micro-spheres or microcylinders with plane waves. In addition, compared with dielectric mi-crospheres or microcylinders, the gold film dielectric microdisk structure can adjust the characteristics of PNJ by changing the refractive index and thickness of dielectric microdisk. Therefore, the proposed dielectric microdisk can provide a new idea for further expanding the generation path of PNJ and conveniently tuning the character-istics of PNJ.
- It should be pointed out that this work is theoretical and predictive, whereas no experimental implementation of it has so far been demonstrated. It is also necessary to specify the parameters of the exciting radiation. The manuscript only gives the wavelength (800 nm), but other key parameters need to be provided, such as the spectral width, divergence, &c.
Answer:Thanks for your careful review and guidance of this paper. Your question has been modified in the paper:
The illuminating wave is a Z-axis polarized plane wave with a wavelength λ=800nm, propagating along the Y-axis with an initial amplitude of 1. The plane wave is irradi-ated onto the grating for coupling, resulting in plasma resonance, and the generated spp SPP propagates along the x-axis, producing PNJ with different properties.
Round 2
Reviewer 1 Report
I should note that the manuscript is greatly improved. However, in the introduction the authors should discuss the works I noted earlier: (1) D. Ju et al. [https://doi.org/10.1063/1.4802958] in which PNJ generation by the SPP illumination of a dielectric disc is proposed for the first time, and (2) I. Minin et al. [https://doi.org/10.1364/OL.391861] where the first experimental demonstration of the plasmonic nanojet production by a microcuboid deposited on a gold film is reported.
Author Response
Thank you for your suggestion. In this paper, we discuss the works suggested by reviewer.
According to your advice, we discuss the works suggested by reviewer in introduction. The detailed modifications are as follows (line 58-63): Besides, due to the limited field portion of SPP in metals, it is often difficult to transmit over long distances, which seriously limits its application value, especially in planar photonic devices. For example, D. Ju et al. first reported nanojet effects excited by surface plasmon polariton at the shadow-side surfaces of dielectric microdisks positioned on gold films in 2013. Besides, in 2020, I. Minin first reported the experimental demonstration of the plasmonic nanojet production by a microcuboid deposited on a gold film.

Reviewer 3 Report
In response to my observations, important information was added to the manuscript that made it more interesting and comprehensible. My comments have been fully addressed by the Authors in the revised manuscript, which may be now published.
Author Response
Thank you for your positive remark and suggestions for our work.